# LEARNING TO DISCOVER SPARSE GRAPHICAL MODELS

**Eugene Belilovsky**
INRIA Galen
University of Paris-Saclay, France
`eugene.belilovsky@inria.fr`

**Kyle Kastner**
MILA Lab
University of Montreal, Canada
`kyle.kastner@umontreal.ca`

**Gael Varoquaux**
INRIA Parietal
Saclay, France
`gael.varoquaux@inria.fr`

**Matthew B. Blaschko**
Center for Processing Speech and Images
KU Leuven, Belgium
`matthew.blaschko@esat.kuleuven.be`

## ABSTRACT

We consider structure discovery of undirected graphical models from observational data. Inferring likely structures from few examples is a complex task often requiring the formulation of priors and sophisticated inference procedures. In the setting of Gaussian Graphical Models (GGMs) a popular estimator is a maximum likelihood objective with a penalization on the precision matrix. Adapting this estimator to capture domain-specific knowledge as priors or a new data likelihood requires great effort. In addition, structure recovery is an indirect consequence of the data-fit term. By contrast, it may be easier to generate training samples of data that arise from graphs with the desired structure properties. We propose here to leverage this latter source of information as training data to learn a function mapping from empirical covariance matrices to estimated graph structures. Learning this function brings two benefits: it implicitly models the desired structure or sparsity properties to form suitable priors, and it can be tailored to the specific problem of edge structure discovery, rather than maximizing data likelihood. We apply this framework to several real-world problems in structure discovery and show that it can be competitive to standard approaches such as graphical lasso, at a fraction of the execution speed. We use convolutional neural networks to parametrize our estimators due to the compositional structure of the problem. Experimentally, our learnable graph-discovery method trained on synthetic data generalizes well: identifying relevant edges in real data, completely unknown at training time. We find that on genetics, brain imaging, and simulation data we obtain competitive (and generally superior) performance, compared with analytical methods.

## 1 INTRODUCTION

Probabilistic graphical models provide a powerful framework for describing the dependencies between a set of variables. Many applications infer the structure of a probabilistic graphical model from data to elucidate the relationships between variables. These relationships are often represented by an undirected graphical model also known as a Markov Random Field (MRF). We focus on a common MRF model, Gaussian graphical models (GGMs). GGMs are used in structure-discovery settings for rich data such as neuroimaging, genetics, or finance (Friedman et al., 2008; Ryali et al, 2012; Mohan et al., 2012; Belilovsky et al., 2016). Although multivariate Gaussian distributions are well-behaved, determining likely structures from few examples is a complex task when the data is high dimensional. It requires strong priors, typically a sparsity assumption, or other restrictions on the structure of the graph, which now make the distribution difficult to express analytically and use.

A standard approach to estimating structure with GGMs in high dimensions is based on the classic result that the zeros of a precision matrix correspond to zero partial correlation, a necessary and sufficient condition for conditional independence (Lauritzen, 1996). Assuming only a few conditional dependencies corresponds to a sparsity constraint on the entries of the precision matrix, leading to a combinatorial problem. Many popular approaches to learning GGMs can be seen as leveraging the

$\ell_1$-norm to create convex surrogates to this problem. Meinshausen & Bühlmann (2006) use nodewise $\ell_1$ penalized regressions. Other estimators penalize the precision matrix directly (Cai et al., 2011; Friedman et al., 2008; Ravikumar et al., 2011). The most popular being the graphical lasso

$$f_{glasso}(\hat{\Sigma}) = \arg\min_{\Theta \succ 0} -\log|\Theta| + \text{Tr}(\hat{\Sigma}\Theta) + \lambda\|\Theta\|_1, \tag{1}$$

which can be seen as a penalized maximum-likelihood estimator. Here $\Theta$ and $\hat{\Sigma}$ are the precision and sample covariance matrices, respectively. A large variety of alternative regularization penalties extend the priors of the graphical lasso (Danaher et al., 2014; Ryali et al, 2012; Varoquaux et al., 2010). However, several problems arise in this approach. Constructing novel surrogates for structured-sparsity assumptions on MRF structures is challenging, as a prior needs to be formulated and incorporated into a penalized maximum likelihood objective which then needs an efficient optimization algorithm to be developed, often within a separate research effort. Furthermore, model selection in a penalized maximum likelihood setting is difficult as regularization parameters are often unintuitive.

We propose to learn the estimator. Rather than manually designing a specific graph-estimation procedure, we frame this estimator-engineering problem as a learning problem, selecting a function from a large flexible function class by risk minimization. This allows us to construct a loss function that explicitly aims to recover the edge structure. Indeed, sampling from a distribution of graphs and empirical covariances with desired properties is often possible, even when this distribution is not analytically tractable. As such we can perform empirical risk minimization to select an appropriate function for edge estimation. Such a framework gives more easy control on the assumed level of sparsity (as opposed to graph lasso) and can impose structure on the sampling to shape the expected distribution, while optimizing a desired performance metric.

For particular cases we show that the problem of interest can be solved with a polynomial function, which is learnable with a neural network (Andoni et al., 2014). Motivated by this fact, as well as theoretical and empricial results on learning smooth functions approximating solutions to combinatorial problems (Cohen et al., 2016; Vinyals et al., 2015), we propose to use a particular convolutional neural network as the function class. We train it by sampling small datasets, generated from graphs with the prescribed properties, with a primary focus on sparse graphical models. We estimate from this data small-sample covariance matrices ($n < p$), where $n$ is the number of samples and $p$ is the dimensionality of the data. Then we use them as training data for the neural network (Figure 2) where target labels are indicators of present and absent edges in the underlying GGM. The learned network can then be employed in various real-world structure discovery problems.

In Section 1.1 we review the related work. In Section 2 we formulate the risk minimization view of graph-structure inference and describe how it applies to sparse GGMs. Section 2.3 describes and motivates the deep-learning architecture we chose to use for the sparse GGM problem in this work. In Section 3 we describe the details of how we train an edge estimator for sparse GGMs. We then evaluate its properties extensively on simulation data. Finally, we show that this edge estimator trained only on synthetic data can obtain state of the art performance at inference time on real neuroimaging and genetics problems, while being much faster to execute than other methods.

## 1.1 RELATED WORK

Lopez-Paz et al. (2015) analyze learning functions to identify the structure of directed graphical models in causal inference using estimates of kernel-mean embeddings. As in our work, they demonstrate the use of simulations for training while testing on real data. Unlike our work, they primarily focus on finding the causal direction in two node graphs with many observations.

Our learning architecture is motivated by the recent literature on deep networks. Vinyals et al. (2015) have shown that neural networks can learn approximate solutions to NP-hard combinatorial problems, and the problem of optimal edge recovery in MRFs can be seen as a combinatorial optimization problem. Several recent works have been proposed which show neural architectures for graph input data (Henaff et al., 2015; Duvenaud et al, 2015; Li et al., 2016). These are based on multi layer convolutional networks, as in our work, or multi-step recurrent neural networks. The input in our approach can be viewed as a complete graph, while the ouput a sparse graph, thus none of these are directly applicable. A related use of deep networks to approximate a posterior distribution can be found in Balan et al. (2015). Finally, Gregor & LeCun (2010); Xin et al. (2016) use deep networks to approximate steps of a known sparse recovery algorithm.

Bayesian approaches to structure learning rely on priors on the graph combined with sampling techniques to estimate the posterior of the graph structure. Some approaches make assumptions on the decomposability of the graph (Moghaddam et al., 2009). The G-Wishart distribution is a popular distribution which forms part of a framework for structure inference, and advances have been recently made in efficient sampling (Mohammadi & Wit, 2015). These methods can still be rather slow compared to competing methods, and in the setting of $p > n$ we find they are less powerful.

## 2 METHODS

### 2.1 LEARNING AN APPROXIMATE EDGE ESTIMATION PROCEDURE

We consider MRF edge estimation as a learnable function. Let $\boldsymbol{X} \in \mathbb{R}^{n \times p}$ be a matrix whose $n$ rows are i.i.d. samples $x \sim P(x)$ of dimension $p$. Let $G = (V, E)$ be an undirected and unweighted graph associated with the set of variables in $x$. Let $\mathcal{L} = \{0, 1\}$ and $N_e = \frac{p(p-1)}{2}$ the maximum possible edges in $E$. Let $Y \in \mathcal{L}^{N_e}$ indicate the presence or absence of edges in the edge set $E$ of $G$, namely

$$Y^{ij} = \begin{cases} 0 & x_i \perp x_j | x_{V \setminus i,j} \\ 1 & x_i \not\perp x_j | x_{V \setminus i,j} \end{cases} \tag{2}$$

We define an approximate structure discovery method $g_w(\boldsymbol{X})$, which produces a prediction of the edge structure, $\hat{Y} = g_w(\boldsymbol{X})$, given a set of data $\boldsymbol{X}$. We focus on $\boldsymbol{X}$ drawn from a Gaussian distribution. In this case, the empirical covariance matrix, $\hat{\boldsymbol{\Sigma}}$, is a sufficient statistic of the population covariance and therefore of the conditional dependency structure. We thus express our structure-recovery problem as a function of $\hat{\boldsymbol{\Sigma}}$: $g_w(\boldsymbol{X}) := f_w(\hat{\boldsymbol{\Sigma}})$. $f_w$ is parametrized by $w$ and belongs to the function class $\mathcal{F}$. We note that the graphical lasso in Equation (1) is an $f_w$ for an appropriate choice of $\mathcal{F}$.

This view on the edge estimator now allows us to bring the selection of $f_w$ from the domain of human design to the domain of empirical risk minimization over $\mathcal{F}$. Defining a distribution $\mathbb{P}$ on $\mathbb{R}^{p \times p} \times \mathcal{L}^{N_e}$ such that $(\hat{\boldsymbol{\Sigma}}, Y) \sim \mathbb{P}$, we would like our estimator, $f_w$, to minimize the expected risk

$$R(f) = \mathbb{E}_{(\hat{\boldsymbol{\Sigma}}, Y) \sim \mathbb{P}}[l(f(\hat{\boldsymbol{\Sigma}}), Y)] \tag{3}$$

Here $l : \mathcal{L}^{N_e} \times \mathcal{L}^{N_e} \to \mathbb{R}^+$ is the loss function. For graphical model selection the $0/1$ loss function is the natural error metric to consider (Wang et al., 2010). The estimator with minimum risk is generally not possible to compute as a closed form expression for most interesting choices of $\mathbb{P}$, such as those arising from sparse graphs. In this setting, Eq. (1) achieves the information theoretic optimal recovery rate up to a constant for certain $\mathbb{P}$ corresponding to uniformly sparse graphs with a maximum degree, but only when the optimal $\lambda$ is used and the non-zero precision matrix values are bounded away from zero (Wang et al., 2010; Ravikumar et al., 2011).

The design of the estimator in Equation (1) is not explicitly minimizing this risk functional. Thus modifying the estimator to fit a different class of graphs (e.g. small-world networks) while minimizing $R(f)$ is not obvious. Furthermore, in practical settings the optimal $\lambda$ is unknown and precision matrix entries can be very small. We would prefer to directly minimize the risk functional. Desired structural assumptions on samples from $\mathbb{P}$ on the underlying graph, such as sparsity, may imply that the distribution is not tractable for analytic solutions. Meanwhile, we can often devise a sampling procedure for $\mathbb{P}$ allowing us to select an appropriate function via empirical risk minimization. Thus it is sufficient to define a rich enough $\mathcal{F}$ over which we can minimize the empirical risk over the samples generated, giving us a learning objective over $N$ samples $\{Y_k, \boldsymbol{\Sigma}_k\}_{k=1}^N$ drawn from $\mathbb{P}$: $\min_w \frac{1}{N} \sum_{k=1}^N l(f_w(\hat{\boldsymbol{\Sigma}}_k), Y_k)$. To maintain tractability, we use the standard cross-entropy loss as a convex surrogate, $\hat{l} : \mathbb{R}^{N_e} \times \mathcal{L}^{N_e}$, given by:

$$\hat{l}(f_w(\hat{\boldsymbol{\Sigma}}), Y) = \sum_{i \neq j} \left( Y^{ij} \log(f_w^{ij}(\hat{\boldsymbol{\Sigma}})) + (1 - Y^{ij}) \log(1 - f_w^{ij}(\hat{\boldsymbol{\Sigma}})) \right). \tag{4}$$

We now need to select a sufficiently rich function class for $f_w$ and a method to produce appropriate $(Y, \hat{\boldsymbol{\Sigma}})$ which model our desired data priors. This will allow us to learn a $f_w$ that explicitly attempts to minimize errors in edge discovery.

## 2.2 DISCOVERING SPARSE GAUSSIAN GRAPHICAL MODELS AND BEYOND

We discuss how the described approach can be applied to recover sparse Gaussian graphical models. A typical assumption in many modalities is that the number of edges is sparse. A convenient property of these GGMs is that the precision matrix has a zero value in the $(i, j)$th entry precisely when variables $i$ and $j$ are independent conditioned on all others. Additionally, the precision matrix and partial correlation matrix have the same sparsity pattern, while the partial correlation matrix has normalized entries.

We propose to simulate our *a priori* assumptions of sparsity and Gaussianity to learn $f_w(\hat{\boldsymbol{\Sigma}})$, which can then produce predictions of edges from the input data. We model $P(x|G)$ as arising from a sparse prior on the graph $G$ and correspondingly the entries of the precision matrix $\boldsymbol{\Theta}$. To obtain a single sample of $\boldsymbol{X}$ corresponds to $n$ i.i.d. samples from $\mathcal{N}(0, \boldsymbol{\Theta}^{-1})$. We can now train $f_w(\hat{\boldsymbol{\Sigma}})$ by generating sample pairs $(\hat{\boldsymbol{\Sigma}}, Y)$. At execution time we standardize the input

---

**Algorithm 1** Training a GGM edge estimator

**for** $i \in \{1, .., N\}$ **do**
    Sample $G_i \sim \mathbb{P}(G)$
    Sample $\boldsymbol{\Sigma}_i \sim \mathbb{P}(\boldsymbol{\Sigma}|G = G_i)$
    $\boldsymbol{X}_i \leftarrow \{x_j \sim N(0, \boldsymbol{\Sigma}_i)\}_{j=1}^n$
    Construct $(Y_i, \hat{\boldsymbol{\Sigma}}_i)$ pair from $(G_i, \boldsymbol{X_i})$
**end for**
Select Function Class $\mathcal{F}$ (e.g. CNN)
Optimize: $\min_{f \in \mathcal{F}} \frac{1}{N} \sum_{k=1}^N \hat{l}(f(\hat{\boldsymbol{\Sigma}}_k), Y_k))$

---

data and compute the covariance matrix before evaluating $f_w(\hat{\boldsymbol{\Sigma}})$. The process of learning $f_w$ for the sparse GGM is given in Algorithm 1. A weakly-informative sparsity prior is one where each edge is equally likely with small probability, versus structured sparsity where edges have specific configurations. For obtaining the training samples $(\hat{\boldsymbol{\Sigma}}, Y)$ in this case we would like to create a sparse precision matrix, $\boldsymbol{\Theta}$, with the desired number of zero entries distributed uniformly. One strategy to do this and assure the precision matrices lie in the positive definite cone is to first construct an upper triangular sparse matrix and then multiply it by its transpose. This process is described in detail in the experimental section. Alternatively, an MCMC based G-Wishart distribution sampler can be employed if specific structures of the graph are desired (Lenkoski, 2013).

The sparsity patterns in real data are often not uniformly distributed. Many real world networks have a small-world structure: graphs that are sparse and yet have a comparatively short average distance between nodes. These transport properties often hinge on a small number of high-degree nodes called hubs. Normally, such structural patterns require sophisticated adaptation when applying estimators like Eq. (1). Indeed, high-degree nodes break the small-sample, sparse-recovery properties of $\ell_1$-penalized estimators (Ravikumar et al., 2011). In our framework such structural assumptions appear as a prior that can be learned offline during training of the prediction function. Similarly priors on other distributions such as general exponential families can be more easily integrated. As the structure discovery model can be trained offline, even a slow sampling procedure may suffice.

## 2.3 NEURAL NETWORK GRAPH ESTIMATOR

In this work we propose to use a neural network as our function $f_w$. To motivate this let us consider the extreme case when $n \gg p$. In this case $\hat{\boldsymbol{\Sigma}} \approx \boldsymbol{\Sigma}$ and thus entries of $\hat{\boldsymbol{\Sigma}}^{-1}$ or the partial correlation that are almost equal to zero can give the edge structure.

**Definition 1** ($\mathbb{P}$-consistency). *A function class $\mathcal{F}$ is $\mathbb{P}$-consistent if $\exists f \in \mathcal{F}$ such that $\mathbb{E}_{(\hat{\boldsymbol{\Sigma}}, Y) \sim \mathbb{P}}[l(f(\hat{\boldsymbol{\Sigma}}), Y)] \rightarrow 0$ as $n \rightarrow \infty$ with high probability.*

**Proposition 1** (Existence of $\mathbb{P}$-consistent neural network graph estimator). *There exists a feed forward neural network function class $\mathcal{F}$ that is $\mathbb{P}$-consistent.*

*Proof.* If the data is standardized, each entry of $\boldsymbol{\Sigma}$ corresponds to the correlation $\rho_{i,j}$. The partial correlation of edge $(i, j)$ conditioned on nodes $\boldsymbol{Z}$, is given recursively as

$$\rho_{i,j|\boldsymbol{Z}} = (\rho_{i,j|\boldsymbol{Z} \setminus z_o} - \rho_{i,z_o|\boldsymbol{Z} \setminus z_o} \rho_{j,z_o|\boldsymbol{Z} \setminus z_o}) \frac{1}{D}. \tag{5}$$

We may ignore the denominator, $D$, as we are interested in $\mathbb{I}(\rho_{i,j|\boldsymbol{Z}} = 0)$. Thus we are left with a recursive formula that yields a high degree polynomial. From Andoni et al. (2014, Theorem 3.1) using gradient descent, a neural network with only two layers can learn a polynomial function of degree $d$ to arbitrary precision given sufficient hidden units. $\square$

**Remark 1.** *Naïvely the polynomial from the recursive definition of partial correlation is of degree bounded by $2^{p-2}$. In the worst case, this would seem to imply that we would need an exponentially*

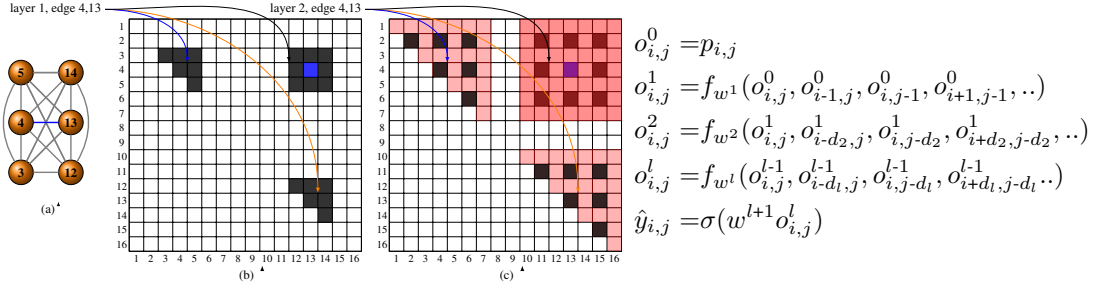

Figure 1: (a) Illustration of nodes and edges "seen" at edge 4,13 in layer 1 and (b) Receptive field at layer 1. All entries in grey show the $o^0_{i,j}$ in covariance matrix used to compute $o^1_{4,13}$. (c) shows the dilation process and receptive field (red) at higher layers

*growing number of hidden nodes to approximate it. However, this problem has a great deal of structure that can allow efficient approximation. Firstly, higher order monomials will go to zero quickly with a uniform prior on $\rho_{i,j}$, which takes values between 0 and 1, suggesting that in many cases a concentration bound exists that guarantees non-exponential growth. Furthermore, the existence result is shown already for a shallow network, and we expect a logarithmic decrease in the number of parameters to peform function estimation with a deep network (Cohen et al., 2016).*

Moreover, there are a great deal of redundant computations in Eq. (5) and an efficient dynamic programming implementation can yield polynomial computation time and require only low order polynomial computations with appropriate storage of previous computation. Similarly we would like to design a network that would have capacity to re-use computations across edges and approximate low order polynomials. We also observe that the conditional independence of nodes $i, j$ given $\boldsymbol{Z}$ can be computed equivalently in many ways by considering many paths through the nodes $\boldsymbol{Z}$. Thus we can choose any valid ordering for traversing the nodes starting from a given edge.

We propose a series of shared operations at each edge. We consider a feedforward network where each edge $i, j$ is associated with a fixed sized vector, $o^k_{i,j}$, of dimensionality $d$ at each layer, $k > 0$. $o^0_{i,j}$ is initialized to the covariance entries at $k = 0$. For each edge we start with a neighborhood of the 6 adjacent nodes, $i, j, i\text{-}1, i\text{+}1, j\text{-}1, j\text{+}1$ for which we take all corresponding edge values from the covariance matrix illustrated in Figure 1. We proceed at each layer to increase the nodes considered for each edge, the output at each layer progressively increasing the receptive field making sure all values associated with the considered nodes are present. The receptive field here refers to the original covariance entries which are accessible by a given, $o^k_{i,j}$ (Luo et al., 2010). The equations defining the process are shown in Figure 1. Here a neural network $f_{w^k}$ is applied at each edge at each layer and a dilation sequence $d_k$ is used. We call a network of this topology a D-Net of depth $l$. We use dilation here to allow the receptive field to grow fast, so the network does not need a great deal of layers. We make the following observations:

**Proposition 2.** *For general $\mathbb{P}$ it is a necessary condition for $\mathbb{P}$-consistency that the receptive field of D-Net covers all entries of the covariance, $\hat{\boldsymbol{\Sigma}}$, at any edge it is applied.*

*Proof.* Consider nodes $i$ and $j$ and a chain graph such that $i$ and $j$ are adjacent to each other in the matrix but are at the terminal nodes of the chain graph. One would need to consider all other variables to be able to explain away the correlation. Alternatively we can see this directly from expanding Eq. (5). □

**Proposition 3.** *A $p \times p$ matrix $\hat{\boldsymbol{\Sigma}}$ will be covered by the receptive field for a D-Net of depth $\log_2(p)$ and $d_k = 2^{k-1}$*

*Proof.* The receptive field of a D-Net with dilation sequence $d_k = 2^{k-1}$ of depth $l$ is $O(2^l)$. We can see this as $o^k_{i,j}$ will receive input from $o^{k-1}_{a,b}$ at the edge of it's receptive field, effectively doubling it. It now follows that we need at least $\log_2(p)$ layers to cover the receptive field. □

Intuitively adjacent edges have a high overlap in their receptive fields and can easily share information about the non-overlapping components. This is analogous to a parametrized message passing. For example if edge $(i, j)$ is explained by node $k$, as $k$ enters the receptive field of edge $(i, j - 1)$,

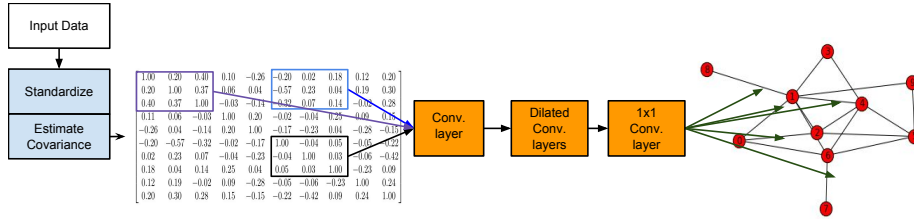

Figure 2: Diagram of the DeepGraph structure discovery architecture used in this work. The input is first standardized and then the sample covariance matrix is estimated. A neural network consisting of multiple dilated convolutions and a final $1 \times 1$ convolution layer is used to predict edges corresponding to non-zero entries in the precision matrix.

the path through $(i, j)$ can already be discounted. In terms of Eq. 5 this can correspond to storing computations that can be used by neighbor edges from lower levels in the recursion.

Here $f_{w^k}$ is shared amongst all nodes and thus we can implement this as a special kind of convolutional network. We make sure that to have considered all edges relevant to the current set of nodes in the receptive field which requires us to add values from filters applied at the diagonal to all edges. In Figure 1 we illustrate the nodes and receptive field considered with respect to the covariance matrix. This also motivates a straightforward implementation using 2D convolutions (adding separate convolutions at $i, i$ and $j, j$ to each $i, j$ at each layer to achieve the specific input pattern described) shown in (Figure 2).

Ultimately our choice of architecture that has shared computations and multiple layers is highly scalable as compared with a naive fully connected approach and allows leveraging existing optimized 2-D convolutions. In preliminary work we have also considered fully connected layers but this proved to be much less efficient in terms of storage and scalability than using deep convolutional networks.

Considering the general $n \gg p$ case is illustrative. However, the main advantages of making the computations differentiable and learned from data is that we can take advantage of the sparsity and structure assumptions on the target function to obtain more efficient results than naive computation of partial correlation or matrix inversion. As $n$ decreases our estimate of $\hat{\rho}_{i,j}$ becomes inexact and here a data driven model which can take advantage of the assumptions on the underlying distribution can more accurately recover the graph structure.

The convolution structure is dependent on the order of the variables used to build the covariance matrix, which is arbitrary. Permuting the input data we can obtain another estimate of the output. In the experiments, we leverage these various estimate in an ensembling approach, averaging the results of several permutations of input. We observe that this generally yields a modest increase in accuracy, but that even a single node ordering can show substantially improved performance over competing methods in the literature.

## 3 EXPERIMENTS

Our experimental evaluations focus on the challenging high dimensional settings in which $p > n$ and consider both synthetic data and real data from genetics and neuroimaging. In our experiments we explore how well networks trained on parametric samples generalize, both to unseen synthetic data and to several real world problems. In order to highlight the generality of the learned networks, we apply the same network to multiple domains. We train networks taking in 39, 50, and 500 node graphs. The former sizes are chosen based on the real data we consider in subsequent sections. We refer to these networks as DeepGraph-39, 50, and 500. In all cases we have 50 feature maps of $3 \times 3$ kernels. The 39 and 50 node network with 6 convolutional layers and $d_k = k + 1$. For the 500 node network with 8 convolutional layers and $d_k = 2^{k+1}$. We use ReLU activations. The last layer has $1 \times 1$ convolution and a sigmoid outputing a value of 0 to 1 for each edge.

We sample $P(X|G)$ with a sparse prior on $P(G)$ as follows. We first construct a lower diagonal matrix, $L$, where each entry has $\alpha$ probability of being zero. Non-zero entries are set uniformly between $-c$ and $c$. Multiplying $LL^T$ gives a sparse positive definite precision matrix, $\mathbf{\Theta}$. This gives us our $P(\mathbf{\Theta}|G)$ with a sparse prior on $P(G)$. We sample from the Gaussian $\mathcal{N}(0, \mathbf{\Theta}^{-1})$ to obtain

samples of $X$. Here $\alpha$ corresponds approximately to a specific sparsity level in the final precision matrix, which we set to produce matrices $92 - 96\%$ sparse and $c$ chosen so that partial correlations range 0 to 1.

Each network is trained continously with new samples generated until the validation error saturates. For a given precision matrix we generate 5 possible $X$ samples to be used as training data, with a total of approximately $100K$ training samples used for each network. The networks are optimized using ADAM (Kingma & Ba, 2015) coupled with cross-entropy loss as the objective function (cf. Sec. 2.1). We use batch normalization at each layer. Additionally, we found that using the absolute value of the true partial correlations as labels, instead of hard binary labels, improves results.

**Synthetic Data Evaluation**   To understand the properties of our learned networks, we evaluated them on different synthetic data than the ones they were trained on. More specifically, we used a completely different third party sampler so as to avoid any contamination. We use DeepGraph-39 on a variety of settings. The same trained network is utilized in the subsequent neuroimaging evaluations as well. DeepGraph-500 is also used to evaluate larger graphs.

We used the `BDGraph` R-package to produce sparse precision matrices based on the G-Wishart distribution (Mohammadi & Wit, 2015) as well as the R-package `rags2ridges` (Peeters et al., 2015) to generate data from small-world networks corresponding to the Watts–Strogatz model (Watts & Strogatz, 1998). We compared our learned estimator against the `scikit-learn` (Pedregosa et al, 2011) implementation of Graphical Lasso with regularizer chosen by cross-validation as well as the Birth-Death Rate MCMC (BDMCMC) method from Mohammadi & Wit (2015).

For each scenario we repeat the experiment for 100 different graphs and small sample observations showing the average area under the ROC curve (AUC), precision@k corresponding to $5\%$ of possible edges, and calibration error (CE) (Mohammadi & Wit, 2015).

For graphical lasso we use the partial correlations to indicate confidence in edges; `BDGraph` automatically returns posterior probabilities as does our method. Finally to understand the effect of the regularization parameter we additionally report the result of graphical lasso under optimal regularizer setting on the testing data.

Our method dominates all other approaches in all cases with $p > n$ (which also corresponds to the training regime). For the case of random Gaussian graphs with n=35 (as in our training data), and graph sparsity of $95\%$, we have superior performance and can further improve on this by averaging permutations. Next we apply the method to a less straightforward synthetic data, with distributions typical of many applications. We found that, compared to baseline methods, our network performs particularly well with high-degree nodes and when the distribution becomes non-normal. In particular our method performs well on the relevant metrics with small-world networks, a very common family of graphs in real-world data, obtaining superior precision at the primary levels of interest. Figure 3 shows examples of random and Watts-Strogatz small-world graphs used in these experiments.

Training a new network for each number of samples can pose difficulties with our proposed method. Thus we evaluted how robust the network DeepGraph-39 is to input covariances obtained from fewer or more samples. We find that overall the performance is quite good even when lowering the number of samples to $n = 15$, we obtain superior performance to the other approaches (Table 1). We also applied DeepGraph-39 on data from a multivariate generalization of the Laplace distribution (Gómez et al., 1998). As in other experiments precision matrices were sampled from the G-Wishart at a sparsity of $95\%$. Gómez et al. (1998, Proposition 3.1) was applied to produce samples. We find that DeepGraph-39 performs competitively, despite the discrepancy between train and test distributions. Experiments with variable sparsity are considered in the supplementary material, which find that for very sparse graphs, the networks remain robust in performance, while for increased density performance degrades but remains competitive.

Using the small-world network data generator (Peeters et al., 2015), we demonstrate that we can update the generic sparse prior to a structured one. We re-train DeepGraph-39 using only 1000 examples of small-world graphs mixed with 1000 examples from the original uniform sparsity model. We perform just one epoch of training and observe markedly improved performance on this test case as seen in the last row of Table 1.

For our final scenario we consider the very challenging setting with 500 nodes and only $n = 50$ samples. We note that the MCMC based method fails to converge at this scale, while graphical lasso is very slow as seen in the timing performance and barely performs better than chance. Our method convincingly outperforms graphical lasso in this scenario. Here we additionally report precision at just the first $0.05\%$ of edges since competitors perform nearly at chance at the $5\%$ level.

| Experimental Setup | Method | Prec@5% | AUC | CE |
|---|---|---|---|---|
| Gaussian Random Graphs ($n = 35, p = 39$) | Glasso | $0.361 \pm 0.011$ | $0.624 \pm 0.006$ | 0.07 |
| | Glasso (optimal) | $0.384 \pm 0.011$ | $0.639 \pm 0.007$ | 0.07 |
| | BDGraph | $0.441 \pm 0.011$ | $0.715 \pm 0.007$ | 0.28 |
| | DeepGraph-39 | $0.463 \pm 0.009$ | $0.738 \pm 0.006$ | 0.07 |
| | DeepGraph-39+Perm | $\mathbf{0.487 \pm 0.010}$ | $\mathbf{0.740 \pm 0.007}$ | 0.07 |
| Gaussian Random Graphs ($n = 100, p = 39$) | Glasso | $0.539 \pm 0.014$ | $0.696 \pm 0.006$ | 0.07 |
| | Glasso (optimal) | $0.571 \pm 0.011$ | $0.704 \pm 0.006$ | 0.07 |
| | BDGraph | $\mathbf{0.648 \pm 0.012}$ | $\mathbf{0.776 \pm 0.007}$ | 0.16 |
| | DeepGraph-39 | $0.567 \pm 0.009$ | $0.759 \pm 0.006$ | 0.07 |
| | DeepGraph-39+Perm | $0.581 \pm 0.008$ | $0.771 \pm 0.006$ | 0.07 |
| Gaussian Random Graphs ($n = 15, p = 39$) | Glasso | $0.233 \pm 0.010$ | $0.566 \pm 0.004$ | 0.07 |
| | Glasso (optimal) | $0.263 \pm 0.010$ | $0.578 \pm 0.004$ | 0.07 |
| | BDGraph | $0.261 \pm 0.009$ | $0.630 \pm 0.007$ | 0.41 |
| | DeepGraph-39 | $0.326 \pm 0.009$ | $0.664 \pm 0.008$ | 0.08 |
| | DeepGraph-39+Perm | $\mathbf{0.360 \pm 0.010}$ | $\mathbf{0.672 \pm 0.008}$ | 0.08 |
| Laplacian Random Graphs ($n = 35, p = 39$) | Glasso | $0.312 \pm 0.012$ | $0.605 \pm 0.006$ | 0.07 |
| | Glasso (optimal) | $0.337 \pm 0.011$ | $0.622 \pm 0.006$ | 0.07 |
| | BDGraph | $0.298 \pm 0.009$ | $0.687 \pm 0.007$ | 0.36 |
| | DeepGraph-39 | $0.415 \pm 0.010$ | $0.711 \pm 0.007$ | 0.07 |
| | DeepGraph-39+Perm | $\mathbf{0.445 \pm 0.011}$ | $\mathbf{0.717 \pm 0.007}$ | 0.07 |
| Gaussian Small-World Graphs (n=35,p=39) | Glasso | $0.387 \pm 0.012$ | $0.588 \pm 0.004$ | 0.11 |
| | Glasso (optimal) | $0.453 \pm 0.008$ | $0.640 \pm 0.004$ | 0.11 |
| | BDGraph | $0.428 \pm 0.007$ | $0.691 \pm 0.003$ | 0.17 |
| | DeepGraph-39 | $\mathbf{0.479 \pm 0.007}$ | $0.709 \pm 0.003$ | 0.11 |
| | DeepGraph-39+Perm | $0.453 \pm 0.007$ | $\mathbf{0.712 \pm 0.003}$ | 0.11 |
| | DeepGraph-39+update | $\mathbf{0.560 \pm 0.008}$ | $\mathbf{0.821 \pm 0.002}$ | 0.11 |
| | DeepGraph-39+update+Perm | $0.555 \pm 0.007$ | $0.805 \pm 0.003$ | 0.11 |

Table 1: For each case we generate 100 sparse graphs with 39 nodes and data matrices sampled (with $n$ samples) from distributions with those underlying graphs. DeepGraph outperforms other methods in terms of AP, AUC, and precision at $5\%$ (the approximate true sparsity). In terms of precision and AUC DeepGraph has better performance in all cases except $n > p$.

We compute the average execution time of our method compared to Graph Lasso and BDGraph on a CPU in Table 4. We note that we use a production quality version of graph lasso (Pedregosa et al, 2011), whereas we have not optimized the network execution, for which known strategies may be applied (Denton et al., 2014).

| Experimental Setup | Method | Prec@0.05% | Prec@5% | AUC | CE |
|---|---|---|---|---|---|
| Gaussian Random Graphs (n=50,p=500) | random | $0.052 \pm 0.002$ | $0.053 \pm 0.000$ | $0.500 \pm 0.000$ | 0.05 |
| | Glasso | $0.156 \pm 0.010$ | $0.055 \pm 0.001$ | $0.501 \pm 0.000$ | 0.05 |
| | Glasso (optimal) | $0.162 \pm 0.010$ | $0.055 \pm 0.001$ | $0.501 \pm 0.000$ | 0.05 |
| | DeepGraph-500 | $0.449 \pm 0.018$ | $0.109 \pm 0.002$ | $0.543 \pm 0.002$ | 0.06 |
| | DeepGraph-500+Perm | $\mathbf{0.583 \pm 0.018}$ | $\mathbf{0.116 \pm 0.002}$ | $\mathbf{0.547 \pm 0.002}$ | $\mathbf{0.06}$ |

Table 2: Experiment on 500 node graphs with only 50 samples repeated 100 times. Improved performance in all metrics.

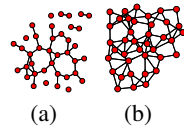

(a)  (b)

Figure 3: Example of (a) random and (b) small world used in experiments

**Cancer Genome Data**    We perform experiments on a gene expression dataset described in Honorio et al. (2012). The data come from a cancer genome atlas from 2360 subjects for various types of cancer. We used the first 50 genes from Honorio et al. (2012, Appendix C.2) of commonly regulated genes in cancer. We evaluated on two groups of subjects, one with breast invasive carcinoma (BRCA) consisting of 590 subjects and the other colon adenocarcinoma (CODA) consisting of 174 subjects.

Evaluating edge selection in real-world data is challenging. We use the following methodology: for each method we select the top-$k$ ranked edges, recomputing the maximum likelihood precision matrix with support given by the corresponding edge selection method. We then evaluate the likelihood on a held-out set of data. We repeat this procedure for a range of $k$. We rely on Algorithm 0 in Hara & Takemura (2010) to compute the maximum likelihood precision given a support. The experiment is repeated for each of CODA and BRCA subject groups 150 times. Results are shown in Figure 4. In all cases we use 40 samples for edge selection and precision estimation. We compare with graphical lasso as well as the Ledoit-Wolf shrinkage estimator (Ledoit & Wolf, 2004). We additionally consider the MCMC based approach described in previous section. For graphical lasso and Ledoit-Wolf, edge selection is based on thresholding partial correlation (Balmand & Dalalyan, 2016).

Additionally, we evaluate the stability of the solutions provided by the various methods. In several applications a low variance on the estimate of the edge set is important. On Table 3, we report

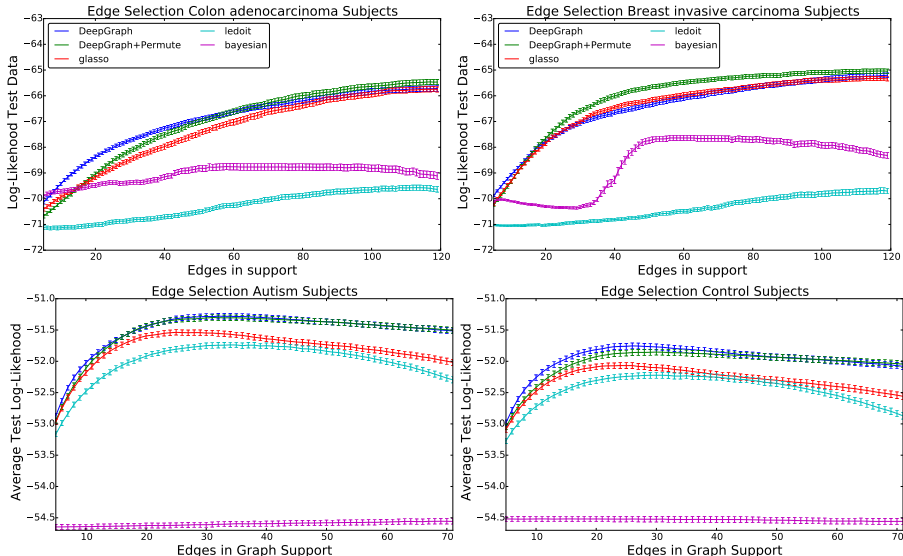

Figure 4: Average test likelihood for COAD and BRCA subject groups in gene data and neuroimaging data using different number of selected edges. Each experiment is repeated 50 times for genetics data. It is repeated approximately 1500 times in the fMRI to obtain significant results due high variance in the data. DeepGraph with averaged permutation dominates in all cases for genetics data, while DeepGraph+Permutation is superior or equal to competing methods in the fMRI data.

Spearman correlations between pairs of solutions, as it is a measure of a monotone link between two variables. DeepGraph has far better stability in the genome experiments and is competitive in the fMRI data.

**Resting State Functional Connectivity**    We evaluate our graph discovery method to study brain functional connectivity in resting-state fMRI data. Correlations in brain activity measured via fMRI reveal functional interactions between remote brain regions. These are an important measure to study psychiatric diseases that have no known anatomical support. Typical connectome analysis describes each subject or group by a GGM measuring functional connectivity between a set of regions (Varoquaux & Craddock, 2013). We use the ABIDE dataset (Di Martino et al, 2014), a large scale resting state fMRI dataset. It gathers brain scans from 539 individuals suffering from autism spectrum disorder and 573 controls over 16 sites.[1] For our experiments we use an atlas with 39 regions of interest derived in Varoquaux et al. (2011).

| | Gene BRCA | Gene COAD | ABIDE Control | ABIDE Autistic |
|---|---|---|---|---|
| Graph Lasso | $0.25 \pm .003$ | $0.34 \pm 0.004$ | $0.21 \pm .003$ | $\mathbf{0.21 \pm .003}$ |
| Ledoit-Wolfe | $0.12 \pm 0.002$ | $0.15 \pm 0.003$ | $0.13 \pm .003$ | $0.13 \pm .003$ |
| Bdgraph | $0.07 \pm 0.002$ | $0.08 \pm 0.002$ | $N/A$ | $N/A$ |
| DeepGraph | $\mathbf{0.48 \pm 0.004}$ | $\mathbf{0.57 \pm 0.005}$ | $\mathbf{0.23 \pm .004}$ | $0.17 \pm .003$ |
| DeepGraph +Permute | $0.42 \pm 0.003$ | $0.52 \pm 0.006$ | $0.19 \pm .004$ | $0.14 \pm .004$ |

Table 3: Average Spearman correlation results for real data showing stability of solution amongst 50 trials

| | 50 nodes (s) | 500 nodes (s) |
|---|---|---|
| sklearn GraphLassoCV | 4.81 | 554.7 |
| BDgraph | 42.13 | N/A |
| DeepGraph | *0.27* | *5.6* |

Table 4: Avg. execution time over 10 trials for 50 and 500 node problem on a CPU for Graph Lasso, BDMCMC, and DeepGraph

We use the network DeepGraph-39, the same network and parameters from synthetic experiments, using the same evaluation protocol as used in the genomic data. For both control and autism patients we use time series from 35 random subjects to estimate edges and corresponding precision matrices. We find that for both the Autism and Control group we can obtain edge selection comparable to graph lasso for very few selected edges. When the number of selected edges is in the range above 25 we begin to perform significantly better in edge selection as seen in Fig. 4. We evaluated stability of the results as shown in Tab. 3. DeepGraph outperformed the other methods across the board.

ABIDE has high variability across sites and subjects. As a result, to resolve differences between approaches, we needed to perform 1000 folds to obtain well-separated error bars. We found that the birth-death MCMC method took very long to converge on this data, moreover the need for many folds to obtain significant results amongst the methods made this approach prohibitively slow to evaluate.

---

[1] http://preprocessed-connectomes-project.github.io/abide/

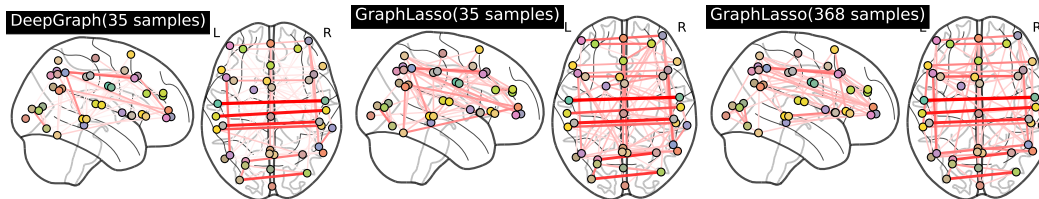

Figure 5: Example solution from DeepGraph and Graph Lasso in the small sample regime on the same 35 samples, along with a larger sample solution of Graph Lasso for reference. DeepGraph is able to extract similar key edges as graphical lasso

We show the edges returned by Graph Lasso and DeepGraph for a sample from 35 subjects (Fig. 5) in the control group. We also show the result of a large-sample result based on 368 subjects from graphical lasso. In visual evaluation of the edges returned by DeepGraph we find that they closely align with results from a large-sample estimation procedure. Furthermore we can see several edges in the subsample which were particularly strongly activated in both methods.

## 4 DISCUSSION AND CONCLUSIONS

Our method was competitive with strong baselines. Even in cases that deviate from standard GGM sparsity assumptions (e.g. Laplacians, small-world) it performed substantially better. When fine-tuning on the target distribution performance further improves. Most importantly the learned estimator generalizes well to real data finding relevant stable edges. We also observed that the learned estimators generalize to variations not seen at training time (e.g. different $n$ or sparsity), which points to this potentialy learning generic computations. This also shows potential to more easily scale the method to different graph sizes. One could consider transfer learning, where a network for one size of data is used as a starting point to learn a network working on larger dimension data.

Penalized maximum likelihood can provide performance guarantees under restrictive assumptions on the form of the distribution and not considering the regularization path. In the proposed method one could obtain empirical bounds under the prescribed data distribution. Additionally, at execution time the speed of the approach can allow for re-sampling based uncertainty estimates and efficient model selection (e.g. cross-validation) amongst several trained estimators.

We have introduced the concept of learning an estimator for determining the structure of an undirected graphical model. A network architecture and sampling procedure for learning such an estimator for the case of sparse GGMs was proposed. We obtained competitive results on synthetic data with various underlying distributions, as well as on challenging real-world data. Empirical results show that our method works particularly well compared to other approaches for small-world networks, an important class of graphs common in real-world domains. We have shown that neural networks can obtain improved results over various statistical methods on real datasets, despite being trained with samples from parametric distributions. Our approach enables straightforward specifications of new priors and opens new directions in efficient graphical structure discovery from few examples.

## ACKNOWLEDGEMENTS

This work is partially funded by Internal Funds KU Leuven, FP7-MC-CIG 334380, DIGITEO 2013-0788D - SOPRANO, and ANR-11-BINF-0004 NiConnect. We thank Jean Honorio for providing pre-processed Cancer Genome Data.

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

|  | mean $\|\hat{\Sigma} - \Sigma\|_2^2$ | mean $\|\hat{\Sigma} - \Sigma\|_\infty$ |
|---|---|---|
| Empirical | 0.0267 | 0.543 |
| Graph Lasso | 0.0223 | 0.680 |
| DeepGraph | 0.0232 | 0.673 |

Table 5: Covariance prediction of ABIDE data. Averaged over 50 trials of 35 samples from the ABIDE Control data

| Experimental Setup | Method | Prec@5% | AUC | CE |
|---|---|---|---|---|
|  | Glasso | $0.464 \pm 0.038$ | $0.726 \pm 0.021$ | 0.02 |
|  | Glasso (optimal) | $0.519 \pm 0.035$ | $0.754 \pm 0.019$ | 0.02 |
| Gaussian Random Graphs | BDGraph | $0.587 \pm 0.033$ | $0.811 \pm 0.017$ | 0.15 |
| (n=35,p=39,sparsity=2%) | DeepGraph-39 | $0.590 \pm 0.026$ | $0.810 \pm 0.019$ | 0.03 |
|  | DeepGraph-39+Perm | $0.598 \pm 0.026$ | $0.831 \pm 0.017$ | 0.03 |
|  | Glasso | $0.732 \pm 0.046$ | $0.562 \pm 0.013$ | 0.32 |
|  | Glasso (optimal) | $0.847 \pm 0.029$ | $0.595 \pm 0.011$ | 0.33 |
| Gaussian Random Graphs | BDGraph | $0.861 \pm 0.015$ | $0.654 \pm 0.013$ | 0.33 |
| (n=35,p=39,sparsity=15%) | DeepGraph-39 | $0.678 \pm 0.032$ | $0.643 \pm 0.012$ | 0.33 |
|  | DeepGraph-39+Perm | $0.792 \pm 0.023$ | $0.660 \pm 0.011$ | 0.33 |

Table 6: For each scenario we generate 100 graphs with 39 nodes, and corresponding data matrix sampled from distributions with those underlying graphs. The number of samples is indicated by $n$.

# A   SUPPLEMENTARY EXPERIMENTS

## A.1   PREDICTING COVARIANCE MATRICES

Using our framework it is possible to attempt to directly predict an accurate covariance matrix given a noisy one constructed from few observations. This is a more challenging task than predicting the edges. In this section we show preliminay experiments which given an empirical covariance matrix from few observations attempts to predict a more accurate covariance matrix that takes into account underlying sparse data dependency structure.

One challenge is that outputs of our covariance predictor must be on the positive semidefinite cone, thus we choose to instead predict on the cholesky decompositions, which allows us to always produce positive definite covariances. We train a similar structure to DeepGraph-39 structure modifying the last layer to be fully connected linear layer that predicts on the cholesky decomposition of the true covariance matrices generated by our model with a squared loss.

We evaluate this network using the ABIDE dataset described in Section 3. The ABIDE data has a large number of samples allowing us to obtain a large sample estimate of the covariance and compare it to our estimator as well as graphical lasso and empirical covariance estimators. Using the large sample ABIDE empirical covariance matrix. We find that we can obtain competitive $\ell_2$ and $\ell_\infty$ norm using few samples. We use 403 subjects from the ABIDE Control group each with a recording of $150 - 200$ samples to construct covariance matrix, totaling $77\,330$ samples (some correlated). This acts as our very approximate estimate of the population $\Sigma$. We then evaluate covariance estimation on 35 samples using the empirical covariance estimator, graphical lasso, and DeepGraph trained to output covariance matrices. We repeat the experiment for 50 different subsamples of the data. We see in 5 that the prediction approach can obtain competitive results. In terms of $\ell_2$ graphical lasso performs better, however our estimate is better than empirical covariance estimation and much faster then graphical lasso. In some applications such as robust estimation a fast estimate of the covariance matrix (automatically embedding sparsity assumptions) can be of great use. For $\ell_\infty$ error we see the empirical covariance estimation outperforms graphical lasso and DeepGraph for this dataset, while DeepGraph performs better in terms of this metric.

We note these results are preliminary, as the covariance predicting networks were not heavily optimized, moreover the ABIDE dataset is very noisy even when pre-processed and thus even the large sample covariance estimate may not be accurate. We believe this is an interesting alternate application of our paper.

## A.2   ADDITIONAL SYNTHETIC RESULTS ON SPARSITY

We investigate the affect of sparsity on DeepGraph-39 which has been trained with input that has sparsity $96\% - 92\%$ sparse. We find that DeepGraph performs well at the $2\%$ sparsity level despite not seeing this at training time. At the same time performance begins to degrade for $15\%$ but is still competitive in several categories. The results are shown in Table 6. Future investigation can consider how alternate variation of sparsity at training time will affect these results.

## A.3   APPLICATION OF LARGER NETWORK ON SMALLER INPUT

We perform preliminary investigation of application of a network trained for a larger number of nodes to a smaller set of nodes. Specifically, we consider the breast invasive carcinoma groups gene data. We now take all 175 valid genes from Appendix C.2 of Honorio et al. (2012). We take the network trained on 500 nodes in the synthetic experiments section. We use the same experimental setup as in the gene experiments. The $175 \times 175$

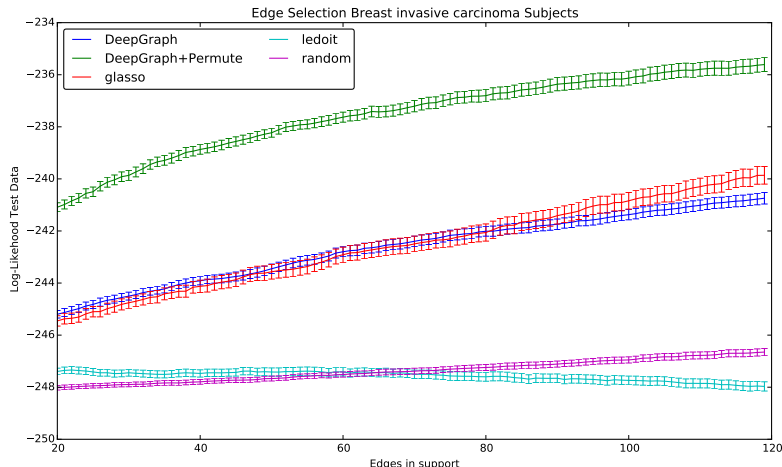

Figure 6: Average test likelihood over 50 trials of applying a network trained for 500 nodes, used on a 175 node problem

covariance matrix from 40 samples and padded to the appropriate size. We observe that DeepGraph has similar performance to graph lasso while permuting the input and ensembling the result gives substantial improvement.

## A.4 PERMUTATION AS ENSEMBLE METHOD

As discussed in Section 2.3, permuting the input and averaging several permutations can produce an improved result empirically. We interpret this as a typical ensembling method. This can be an advantage of the proposed architecture as we are able to easily use standard ensemble techniques. We perform an experiment to further verify that indeed the permutation of the input (and subsequent inverse permutation) allows us to produce separate classifiers that have uncorrelated errors.

We use the setup from the synthetic experiments with DeepGraph-39 in Section 3 with $n = 35$ and $p = 39$. We construct 20 permutation matrices as in the experimental section. Treating each as a separate classifier we compute the correlation coefficient of the errors on 50 synthetic input examples. We find that the average correlation coefficient of the errors of two classifiers is $0.028 \pm 0.002$, suggesting they are uncorrelated. Finally we note the individual errors are relatively small, as can already be inferred from our extensive experimental results in Section 3. We however compute the average absolute error of all the outputs across each permutation for this set of inputs as $0.03$, notably the range of outputs is $0$ to $1$. Thus since prediction error differ at each permutation but are accurate we can average and yield a lower total prediction error.

Finally we note that our method is extremely efficient computationally thus averaging the results of several permutations is practical even as the graph becomes large.

