# Peer review of "Learning to Discover Sparse Graphical Models"

_ICLR 2017 — rejected_

[Official Review · AnonReviewer1 · rating 5 · confidence 2 · 14 Dec 2016]
**Advantage of the proposed method**

This paper proposes a new method for learning graphical models. Combined with a neural network architecture, some sparse edge structure is estimated via sampling methods. In introduction, the authors say that a problem in graphical lasso is model selection. However, the proposed method still implicitly includes model selection. In the proposed method, $P(G)$ is a sparse prior, and should include some hyper-parameters. How do you tune the hyper-parameters? Is this tuning an equivalent problem to model section? Therefore, I do not understand real advantage of this method over previous methods. What is the advantage of the proposed method?

Another concern is that this paper is unorganized. In Algorithm 1, first, G_i and \Sigma_i are sampled, and then x_j is sampled from N(0, \Sigma). Here, what is \Sigma? Is it different from \Sigma_i? Furthermore, how do you construct (Y_i, \hat{\Sigma}_i) from (G_i, X_i )? Finally, I have a simple question: Where is input data X (not sampled data) is used in Algorithm 1?

What is the definition of the receptive field in Proposition 2 and Proposition 3?

[Official Review · AnonReviewer2 · rating 7 · confidence 3 · 16 Dec 2016]
**Interesting algorithm to estimate sparse graph structure**

The paper proposes a novel algorithm to estimate graph structures by using a convolutional neural network to approximate the function that maps from empirical covariance matrix to the sparsity pattern of the graph. Compared with existing approaches, the new algorithm can adapt to different network structures, e.g. small-world networks, better under the same empirical risk minimization framework. Experiments on synthetic and real-world datasets show promising results compared with baselines.

In general, I think it is an interesting and novel paper. The idea of framing structure estimation as a learning problem is especially interesting and may inspire further research on related topics. The advantage of such an approach is that it allows easier adaptation to different network structure properties without designing specific regularization terms as in graph lasso.

The experiment results are also promising. In both synthetic and real-world datasets, the proposed algorithm outperforms other baselines in the small sample region. 

However, the paper can be made clearer in describing the network architectures. For example, in page 5, each o^k_{i,j} is said be a d-dimensional vector. But from the context, it seems o^k_{i,j} is a scalar (from o^0_{i,j} = p_{i,j}). It is not clear what o^k_{i,j} is exactly and what d is. Is it the number of channels for the convolutional filters?

Figure 1 is also quite confusing. Why in (b) the table is 16 x 16 whereas in (a) there are only six nodes? And from the figure, it seems there is only one channel in each layer? What do the black squares represent and why are there three blocks of them. There are some descriptions in the text, but it is still not clear what they mean exactly.

For real-world data, how are the training data (Y, Sigma) generated? Are they generated in the same way as in the synthetic experiments where the entries are uniformly sparse? This is also related to the more general question of how to sample from the distribution P, in the case of real-world data.

[Official Review · AnonReviewer3 · rating 6 · confidence 3 · 20 Dec 2016]

I sincerely apologize for the late-arriving review. 

This paper proposes to frame the problem of structure estimation as a supervised classification problem. The input is an empirical covariance matrix of the observed data, the output the binary decision whether or not two variables share a link. The paper is sufficiently clear, the goals are clear and everything is well described. 

The main interesting point is the empirical results of the experimental section. The approach is simple and performs better than previous non-learning based methods. This observation is interesting and will be of interest in structure discovery problems. 

I rate the specific construction of the supervised learning method as a reasonable attempt attempt to approach this problem. There is not very much technical novelty in this part. E.g., an algorithmic contribution would have been a method that is invariant to data permutation could have been a possible target for a technical contribution. The paper makes no claims on this technical part, as said, the method is well constructed and well executed. 

It is good to precisely state the theoretical parts of a paper, the authors do this well. All results are rather straight-forward, I like that the claims are written down, but there is little surprise in the statements. 

In summary, the paper makes a very interesting observation. Graph estimation can be posed as a supervised learning problem and training data from a separate source is sufficient to learn structure in novel and unseen test data from a new source. Practically this may be relevant, on one hand the empirical results are stronger with this method, on the other hand a practitioner who is interested in structural discovery may have side constraints about interpretability of the deriving method. From the Discussion and Conclusion I understand that the authors consider this as future work. It is a good first step, it could be stronger but also stands on its own already.

[Author Response · Eugene Belilovsky · 12 Jan 2017]
**General Comments**

We thank the reviewers for their comments and for providing us with excellent feedback. We have updated the paper with clarifications in Section 2.3 as well as an Appendix (A.4) with some additional experiments which analyze permuted inputs. We have also made an early release of our code available at

[Final Decision · Program Chairs · 06 Feb 2017]
**ICLR committee final decision**

The authors provide a modern twist to the classical problem of graphical model selection. Traditionally, the sparsity priors to encourage selection of specific structures is hand-engineered. Instead, the authors propose using a neural network to train for these priors. Since graphical models are useful in the small-sample regime, using neural networks directly on the training data is not effective. Instead, the authors propose generating data based on the desired graph structures to train the neural network. 
 
 While this is a nice idea, the paper is not clear and convincing enough to be accepted to the conference, and instead, recommend it to the workshop track.